# Language Profiles of School-Aged Children with 22q11.2 Copy Number Variants

**DOI:** 10.3390/genes14030679

**Published:** 2023-03-09

**Authors:** Jente Verbesselt, Cynthia B. Solot, Ellen Van Den Heuvel, T. Blaine Crowley, Victoria Giunta, Jeroen Breckpot, Donna M. McDonald-McGinn, Inge Zink, Ann Swillen

**Affiliations:** 1Department of Human Genetics, Catholic University Leuven, 3000 Leuven, Belgium; 2Research Group Experimental Oto-Rhino-Laryngology (ExpORL), Department of Neurosciences, Catholic University Leuven, 3000 Leuven, Belgium; 3Center for Childhood Communication, 22q and You Center, Cleft Lip and Palate Program, Children’s Hospital of Philadelphia, Philadelphia, PA 19104, USA; 4Centre for Developmental Disorders, University Hospital Brussels, 1090 Jette, Belgium; 5Children’s Hospital of Philadelphia, Philadelphia, PA 19104, USA; 6Centre for Human Genetics, University Hospitals Leuven, 3000 Leuven, Belgium; 7Perelman School of Medicine, University of Pennsylvania, Philadelphia, PA 19104, USA; 8Department of Human Biology and Medical Genetics, Sapienza University, 00185 Rome, Italy; 9MUCLA, Department of Oto-Rhino-Laryngology, Head & Neck Surgery, University Hospitals Leuven, 3000 Leuven, Belgium

**Keywords:** copy number variants, 22q11.2 deletion syndrome, 22q11.2 duplication, language, communication

## Abstract

Although it is known that copy number variants (CNVs) on chromosome 22, such as 22q11.2 deletion (22q11.2DS) and 22q11.2 duplication (22q11.2Dup) syndromes, are associated with higher risk for neurodevelopmental issues, few studies have examined the language skills across 22q11.2Dup nor compared them with the 22q11.2DS. The current study aims to characterize language abilities in school-aged children with 22q11.2Dup (*n* = 29), compared to age-matched children with 22q11.2DS (*n* = 29). Standardized language tests were administered, assessing receptive and expressive language skills across different language domains. Results indicate that children with 22q11.2Dup demonstrate significantly more language problems compared to the general population. Mean language skills were not significantly different among children with 22q11.2 CNVs in this cohort. While children with 22q11.2DS demonstrated language difficulties starting at the word level, the most common language problems in children with 22q11.2Dup started at the sentence level. Importantly, both expressive and receptive language as well as lexico-semantic and morphosyntactic domains were impaired in children with 22q11.2 CNVs. Early identification, therapeutic intervention, and follow-up of language impairments in children with 22q11.2Dup are recommended to support language development and to reduce longitudinal impact of language and communicative deficits.

## 1. Introduction

Language and speech problems are major features of 22q11.2 deletion syndrome (22q11.2DS), with most children showing communication delays and up to 95% diagnosed with speech-language disorders [1,2,3]. Since the duplication in the same chromosomal region is generally associated with milder phenotypes, one might wonder whether children with 22q11.2 duplication (22q11.2Dup) are less vulnerable to speech and language problems [3,4,5]. 

Until now, little has been reported regarding language in children with 22q11.2Dup. Some case reports have mentioned speech or language delays but, in most instances, these problems were not further specified, and no clear distinction was made between speech and language problems [4,6,7,8,9,10,11,12]. A prospective study using questionnaires compared social-communication skills in 19 children with 22q11.2Dup, 11 unaffected siblings, and 19 children with 22q11.2DS [13]. Parents completed the Children’s Communication Checklist (CCC-2), measuring speech, structural and pragmatic language, and social skills [14,15]. Speech-language delays were found in 79% (15/19) of children with 22q11.2Dup and 95% (18/19) of children with 22q11.2DS. Parents reported general communication problems in 47% of children with 22q11.2Dup (9/19), compared with 79% (15/19) of children with 22q11.2DS. The results also revealed that children with 22q11.2Dup were in an intermediate position between their siblings and children with 22q11.2DS [13]. Another recent study [5] in 28 patients with 22q11.2Dup demonstrated delayed speech and language milestones in 68%. In addition, a subgroup of patients who underwent standardized testing showed language problems and one received a formal diagnosis of developmental language disorder (DLD). Longitudinal language data in six patients revealed a relatively stable trajectory in 3/6, catch-up with peers in 1/6, and a growing-into-deficit profile in 2/6. Growing into deficit means that patients are making insufficient progress with age, resulting in an increasing gap in language skills in relation to their typically developing peers [5,16,17,18]. 

Until now, language has only been evaluated in an indirect way through questionnaires or non-specific language testing. The current study aimed to characterize language profiles through direct standardized assessments in school-aged children with 22q11.2Dup and compare them to the skills of typically developing peers and age-matched children with the 22q11.2DS. To obtain a larger sample and higher statistical power, children seen at two different clinical genetics centers were studied: CME-Leuven in Belgium and Children’s Hospital of Philadelphia (CHOP) in the USA. The following research questions are addressed. 

Were both cohorts (Leuven and Philadelphia) sufficiently comparable to combine the data?

If both cohorts were not statistically significantly different, their data would be combined for the subsequent analyses. 

2.Were language skills of the two CNV groups (22q11.2Dup, 22q11.2DS) comparable to the scores of typically developing peers (norm group scores)?

Language abilities of children in the CNV groups were expected to differ from the normative sample.

3.Which language skills did children with 22q11.2Dup show in relation to age-matched children with 22q11.2DS?

In accordance with previous indirect results (parent-reported), we hypothesized that children with 22q11.2Dup would fall between the general population and age-matched children with 22q11.2DS.

4.Did confounding factors have an impact on the language outcome, such as sex, comorbid ASD, ADHD, inheritance pattern, socioeconomic status (SES), and medical issues such as congenital heart disease (CHD), palatal defects, and hearing loss (HL)?

Confounding factors were expected to have an impact on language outcomes in children with 22q11.2 CNVs.

5.Did genotype–phenotype correlations reveal duplicated regions or genes on 22q11.2 critical for language development?

Genotype–phenotype correlations were expected to reveal critical regions on 22q11.2 for language development.

## 2. Materials and Methods

### 2.1. Participants

A total of 58 school-aged children between 6 and 16 years of age were studied, consisting of 29 unrelated children with 22q11.2Dup and 29 unrelated children with 22q11.2DS. Exclusion criteria included: first language other than Dutch/English, extreme prematurity (i.e., gestational age < 32 weeks), and moderate to severe hearing loss (≥35 dB *HL*) because of the known impact on language outcome [19,20,21,22,23,24]. Additionally, children with CNVs outside of the standard LCR22A-LCR22D region or those children with more than one pathogenic genetic variant were excluded because of the lack of a minimally overlapping region and the impact on the phenotype, respectively. Children with comorbid neurodevelopmental disorders (NDDs) such as autism spectrum disorder (ASD) and attention-deficit/hyperactivity disorder (ADHD) were included, due to their high comorbidity with CNVs. 

All participants with 22q11.2 CNVs had a laboratory confirmed diagnosis based on fluorescence in situ hybridization (FISH), SNP microarray, or multiplex ligation-dependent probe amplification (MLPA). The typical 3 MB deletion/duplication with breakpoints situated at LCR22A-LCR22D was identified in all patients with 22q11.2DS (29) and 16 patients with 22q11.2Dup. Nested duplications included LCR22A-LCR22B (2), LCR22A-LCR22E (2), LCR22A-LCR22H, LCR22B-LCR22C, LCR22B-LCR22F, and LCR22C-LCR22D (3). All duplications with breakpoints at the LCR22A-LCR22B region included the important developmental driver gene *TBX1*. More than half of the 22q11.2Dup were inherited (57%), while most 22q11.2DS occurred as a de novo event (95%). All 58 children with 22q11.2 CNVs were index patients referred for genetic testing due to medical/developmental/behavioral differences. 

Table 1 includes the demographic and clinical characteristics of all participants. Children from CHOP were, on average, 1.9 years younger than children from the Leuven site. This difference was not statistically significantly different for 22q11.2Dup (*t* = −1.776, *p* = 0.087) or 22q11.2DS (*t* = −1.740, *p* = 0.093). Data on developmental milestones and education were retrieved from digital medical records/questionnaires completed by parents. Delayed speech-language milestones were present in 67% of children with 22q11.2Dup and 96% of children with 22q11.2DS. Speech-language therapy was received by 83% of children with 22q11.2Dup and 100% of children with 22q11.2DS. Data on several genetic, environmental, developmental, and medical confounding factors were collected. Parental education (based on the mother’s educational attainment) was used as a marker of SES and classified according to the International Standard Classification of Education (ISCED) of UNESCO [25,26] into three categories: low (primary education or lower grades of high school), middle (graduated from secondary/high school), high (graduated Bachelor, Masters, or Doctor of Philosophy). The presence and severity of congenital heart disease were classified by structural complexity using a three-point scale based on the classification by Billett et al. [27], whereas palatal defects were classified as abnormal when having either a structural and/or functional palatal abnormality such as cleft lip/palate, cleft palate, submucous cleft palate or velopharyngeal dysfunction. Mild hearing loss was defined as having hearing thresholds between 20 and 35 dB *HL* [28]. 

### 2.2. Research Design

Participants were recruited across two sites: the Centre for Human Genetics of UZ Leuven in Belgium and the 22q and You Center at CHOP in the USA. For the CHOP cohort, data were obtained retrospectively through medical records/IRB approved REDCap database. Consequently, the amount of data available varied across different clinical features and patients, resulting in missing data for certain variables such as type of education, SES, speech-language delays, inheritance pattern and formal NDD diagnoses (Table 1). Hence, the total number of patients may vary depending on the available data in the described demographic and clinical features. For Dutch participants, data were prospectively collected during consultations at the hospital or home visits following a standardized research protocol. Further methodological details can be found in the study of Verbesselt et al. [13], in which we have previously reported on a subgroup of these participants. 

This study used a cross-sectional research design with both independent and pairwise comparisons. First, both cohort sites were compared across both CNV groups. Second, 29 chronological age-matched (CA) CNV pairs were compared. Cohort sites were considered by matching only within the same cohort. Age matching was within 0.7 years of age, with a mean difference of 1.72 months. Paired samples Wilcoxon signed-rank *t*-tests confirmed that there were no significant differences for the matching parameter (*W* = 87.500, *p* = 0.617). According to Mervis and Klein-Tasman [29], *p*-values of > 0.50 suggest that group distributions are sufficiently overlapping to be considered properly matched on the matching parameter [30]. 

### 2.3. Measurements

Participants’ language abilities were measured using standardized language instruments: the Dutch adaptation of the Clinical Evaluation of Language Fundamentals-Fourth edition (CELF-4-NL) [31] and/or the CELF-Preschool-Second Edition (CELF-P2-NL) [32]. The CELF-Third, Fourth and Fifth editions (CELF-3, CELF-4 and CELF-5) [33,34,35,36] were used in the English-speaking cohort. The CELF assesses both receptive and expressive language across different language domains (i.e., semantics, syntax, and morphology); it is used in clinical practice to identify patients with language impairments, plan interventions, and evaluate progress over time. In addition, the CELF has normed references from 3–6 years of age on the CELF-P2 and 5–18 (Dutch version) or 5–21 years of age (English version) on the CELF (versions 3–5). Based on the chronological age (CA) of the child, raw scores of each subtest were converted into scaled scores (SS) with a mean of 10 and a standard deviation (SD) of 3. Scaled scores of 7–13 were considered within the average range. Children with scaled scores of ≤6 were considered to have mild–moderate language problems, whereas scaled scores of ≤3 were interpreted as severe language problems. Receptive language subtests included the following: Concepts and Following Directions (CFD), Sentence Structure (SST), Understanding Spoken Paragraphs and Semantic Relationships (SR). Expressive language subtests included: Word Structure (WS), Recalling Sentences (RS), Formulated Sentences (FS), Word Classes (WC), Expressive Vocabulary (EV), Word Definitions (WD), and Sentence Assembly (SA). Core language, receptive, and expressive index scores were calculated based on CA (mean = 100, SD = 15) with a clinical cut-off of 85 (16th percentile, −1 SD) for mild language problems, 77 (6th percentile, −1.5 SD) for moderate language problems and 70 (2nd percentile, −2 SD) for severe language deficits.

The combination of receptive and expressive subtests to obtain the core, or composite, language scores (CLS) differed depending on the chronological age and test edition. In each test, however, the CLS was a measure for overall language ability. A review study found strong correlations between CELF-4 and CELF-5 composite scores [37]. Receptive language index (RLI) and expressive language index (ELI) were calculated to measure language production and comprehension. An additional expressive composite score was calculated based on the subtests Recalling Sentences (RS) and Formulated Sentences (FS), since these were consistently administered in children of all ages, regardless of the specific test or edition. The constructed composite was formed by the combined scaled scores of both subtests, with a mean scaled score of 20: e.g., a child with scaled scores of 6 on FS and 8 on RS has an expressive composite of 14.

### 2.4. Data Analysis

Independent *t*-tests were used to investigate whether differences exist between the Leuven and CHOP cohort. Data were combined in the subsequent analyses once CNV groups were determined to be comparable. Depending on the normality of the sample, Student’s or Wilcoxon signed-rank one-sample *t*-tests were run to determine whether the language skills of the two target groups differ from the normative sample. Cross-CNV comparisons were carried out using paired sample Student’s *t*-tests. In addition, several genetic, environmental, developmental, and medical confounding factors were investigated through exploratory linear mixed models, with the CNV group and each confounding factor separately as fixed effects and intrafamilial pairs as random effect. 

Due to the anticipated heterogeneity within the target CNV groups, quantitative analyses were complemented with qualitative analyses and descriptive data were generated for all variables and groups. A *p*-value < 0.05 was considered statistically significant. Bonferroni correction was applied to reduce type I-errors because of multiple testing. Adapted *p*-values ranged from 0.008 to 0.025. For all outcome variables, 95% confidence intervals were calculated. Cohen’s *d* was calculated as effect size measure for parametric analyses and values of 0.2, 0.5 and 0.8 were, according to Cohen’s guidelines [38,39], interpreted as small, moderate, and large effects, respectively. Analyses were carried out using JASP version 0.16.3 [40] and R 4.2.1 [41,42]. 

## 3. Results

### 3.1. Cohort Site Differences

Figure 1 provides the boxplots of the CELF Core Language Scores (CLS) across cohort sites and CNV groups, with the light and darker gray zones delineating mild–moderate and severe language problems, respectively, and the dotted line referring to the mean in the normative sample. Mean composite scores across both cohort sites and CNV groups are summarized in Table 2. The table illustrates that, on average, children with 22q11.2Dup from CHOP had better language skills compared to the Dutch-speaking children, though independent *t*-tests revealed that the data were not statistically significantly different, as shown in Table 2. In contrast, children with 22q11.2DS from both CHOP and Leuven showed similar language scores. The *p*-values of the 22q11.2DS group met the cut-off of *p* > 0.50 and, therefore, could be considered well matched, while the *p*-value for CLS in the 22q11.2Dup did not meet this criterion [29]. Both cohort sites were not statistically significantly different but not properly matched either in the case of the duplications. Therefore, the subsequent analyses were conducted on the combined Leuven and CHOP cohort and on each cohort site separately. 

### 3.2. 22q11.2 CNVs Compared to Norm Group Scores

Figure 2 displays the boxplots of the combined CELF CLS for each CNV group, with the dotted line pointing to the average of the normative sample and the gray zones showing the cut-offs for mild–moderate to severe core language impairments. A wide range of scores can be observed for both CNV groups. 

Comparative CELF core language scores are summarized in Figure 3. The dotted line represents the normal distribution of the normative sample (mean = 100, SD = 15). Compared to the norm group, negative shifts of 1.41 SD and 1.91 SD were observed in children with 22q11.2Dup and children with 22q11.2DS, respectively. The normative distributions of children with 22q11.2 CNVs show considerable overlap. One-sample Student’s *t*-tests were carried out to determine whether the core language scores of the children with 22q11.2 CNVs differed from the normative sample. Results indicated that children with 22q11.2 CNVs scored statistically significantly lower on language compared to the norm group (*p* < 0.001) with large effect sizes (*d* < −1.441) and the results remained significant after Bonferroni correction.

### 3.3. Quantitative and Qualitative Cross-CNV Comparisons

Mean CELF scores in Table 3 show mild–moderate core language impairments in children with 22q11.2Dup compared with moderate core language impairments in children with 22q11.2DS. Student’s paired *t*-tests were performed to compare across CNVs, revealing no statistically significant differences in CLS between 22q11.2DS and 22q11.2Dup. Similarly, children with 22q11.2DS did not score significantly lower on the constructed expressive composite consisting of the subtests Recalling Sentences and Formulated Sentences. Similar scores were found for receptive (RLI) and expressive language indices (ELI) in children with 22q11.2Dup (mean RLI = 79.04, mean ELI = 77.00; *n* = 27) and in children with 22q11.2DS (mean RLI = 73.33, mean ELI = 71.67; *n* = 12). Subtest scores revealed similar distributions across all subtests, suggesting that children with 22q11.2DS and children with 22q11.2Dup in this sample have comparable language skills. Within the 22q11.2Dup group, children with delayed speech-language milestones in infancy (*n* = 18) demonstrated mean CLS of 73.44 compared to mean CLS of 88.67 in children without speech-language delays (*n* = 9). Independent *t*-tests confirmed that children with delayed milestones in infancy performed statistically significantly lower than children without speech-language delays (*t* = −2.743, *p* = 0.011, *d* = −1.120). 

Proportions of children with difficulties across CELF composite scores and subtests are summarized in Table 4. Based on the cut-off scores, core language impairments were ascertained in 62% of children with 22q11.2Dup and 83% of children with 22q11.2DS. There were severe impairments in 28% of children with 22q11.2Dup and 45% of children with 22q11.2DS. At the subtest level, the most common impairments in children with 22q11.2Dup were issues with Recalling Sentences in 66%, Concepts and Following Directions in 63%, Sentence Comprehension (5.0–8.11) or Semantic Relations (≥9.0) in 54% and Formulated sentences in 52%. In children with 22q11.2DS, the most common difficulties were problems with Word Definitions in 90%, Formulated Sentences in 76%, Concepts and Following Directions in 72%, Word Classes in 58%, Expressive Vocabulary in 57%, Recalling Sentences in 55%, and Sentence Comprehension (5.0–8.11) or Semantic Relations (≥9.0) in 52%. Within the 22q11.2Dup group, most children with delayed speech-language milestones in infancy also showed impaired CLS (15/18), whereas 17% obtained average CLS (3/18). Most children without speech-language delays in infancy also showed average CLS (7/9), while 22% (2/7) showed impaired CLS. Within the 22q11.2DS group, most children with speech-language delays also demonstrated impaired CLS (18/22), whereas 18% (4/22) obtained average CLS. The child without speech-language delays also obtained average CLS. 

### 3.4. Influence of Confounding Factors

Exploratory mixed models were carried out to delineate the impact of sex, comorbid ASD, ADHD, inheritance pattern, SES, and medical issues such as congenital heart disease (CHD), palatal defects, and mild hearing loss on language outcome, while accounting for intrafamilial pairs. Results indicate that none of the additional variables separately were statistically significantly associated with CELF CLS, except for high SES combined with specific CNV (CNV: *t* = 2.511, *p* = 0.0154; middle SES: *t* = 1.988, *p* = 0.0524; high SES: *t* = 2.057, *p* = 0.0229). The best fitted model included two factors in addition to the specific CNV (*t* = 1.785, *p* = 0.0873): the diagnosis of ASD (*t* = −1.452, *p* = 0.1546) and the SES (middle: *t* = 1.593, *p* = 0.1190; high: *t* = 2.169, *p* = 0.0359). Nevertheless, these associations were not statistically significant, except for high SES, although it did not remain significant after Bonferroni correction. 

Descriptive data of scaled language scores were generated for all confounding factors. However, numbers were too imbalanced across groups to make meaningful comparisons and qualitative differences need to be interpreted with caution. Only qualitative differences on sufficiently large (*n* = 5) subgroups were explored. Within the 22q11.2Dup group, similar language scores were found independent of sex (average F: 76.4, M: 80.5), inheritance pattern (average de novo: 77.3, inherited: 79.7), and middle or high SES (average middle: 82.4, high: 80.4). Children with ASD (*n* = 6) performed, on average, five points higher than children without formal diagnosis of ASD (*n* = 23), while children with mild hearing loss (*n* = 5) scored, on average, nine points higher than children without hearing loss (*n* = 24). 

Within the 22q11.2DS group, language scores were comparable regardless of sex (average F: 69.2, M: 73.4) and the presence of mild hearing loss (average HL: 73.5, no HL: 69.2). Children with either structural and/or functional palatal defects (*n* = 22) scored, on average, five points lower than children without palatal defects (*n* = 7). Children with ASD (*n* = 7) performed statistically significantly lower compared to children without formal diagnosis of ASD (*n* = 15, *t* = 4.089, *p* < 0.001, *d* = 0.550). Finally, children with high SES (*n* = 15) scored, on average, eight points higher compared to children with middle SES (*n* = 13, average high: 75.4, middle: 67.3). 

### 3.5. Genotype–Phenotype Correlations

Genotype–phenotype comparisons in typical duplications (LCR22A-LCR22D) versus nested duplications distal to LCR22B reveal that 63% (10/16) children with the typical duplication show considerable language problems compared to 60% (3/5) children with nested duplications without LCR22A-LCR22B. We delineated three minimal critical regions for language impairment: LCR22A-LCR22B in 13 children, LCR22B-LCR22C in 13 children, and LCR22C-LCR22D in 15 children.

## 4. Discussion 

The current study aimed to characterize language skills using standardized language instruments in school-aged children with 22q11.2Dup, in comparison to skills of age-matched children with 22q11.2DS. Since children were studied from two cohort sites, cohort site-related differences were first explored. Results revealed no statistically significant differences between both cohorts for children with 22q11.2Dup or 22q11.2DS. On average, children with 22q11.2Dup from CHOP scored six points higher on CLS compared to children with 22q11.2Dup from Leuven. These results may be partially explained by the fact that children from CHOP were, on average, almost two years younger and that different versions of the same test (CELF-4 in Leuven vs. CELF-5 in CHOP) were used to assess language abilities. Cultural or spoken language differences between both countries may also contribute to these differences. Additionally, since children from CHOP were slightly younger, their normed scores may still decrease with increasing age, reflecting a growing-into-deficit profile, previously reported in a subgroup of the Leuven cohort and in children with 22q11.2DS [1,5]. Finally, higher language outcomes in the CHOP cohort could be related to intellectual functioning in this group, an area requiring further study. Nevertheless, these rather small mean differences were not sufficient to consider groups as not comparable. 

In agreement with the literature [3,43,44,45], we found a slight male predominance in the current cohort of 22q11.2Dup. Percentages for medical issues fell within the range of reported percentages in previous both smaller and larger studies for CHD (0–24%) [3,4,8,43,46,47,48,49], mild hearing loss (4–42%) [3,4,6,8,46,48,49], and palatal defects (8–25%) [8,12,46]. Other studies reported, on average, slightly higher rates of ASD (7–46%) and ADHD (27–44%) [3,6,8,44,45,49,50] and lower rates of language delays (33–54%) [8,12,49]. The current cohort of 22q11.2DS consisted of a relatively high proportion of males, whereas other larger studies reported more even male/female distributions. In addition, the current cohort demonstrated high rates of medical issues (CHD and palatal defects) and elevated rates of NDD (ASD and ADHD), which is in line with other studies [3,18,44,45,51,52]. Based on these findings, the current cohort of children with 22q11.2 CNVs seemed to be representative of what has been described in the literature so far. 

Comparisons to the normative sample confirmed that children with 22q11.2 CNVs have statistically significantly lower language scores in relation to typically developing peers, which is in accordance with results based on parental reports [13]. In addition, a shift of approximately −2 SD in the 22q11.2DS group is consistent with findings on their cognitive profiles in previous studies [53,54,55,56]. We found a shift of approximately −1.5 SD in the 22q11.2Dup group, which is at the lower end of the range of their cognitive capabilities based on previous studies. In particular, Chawner et al. [44] found a downward shift of 0.8 SD with a mean FSIQ of 88 in 32 patients with 22q11.2Dup, whereas Modenato et al. [57,58] reported mean FSIQ of 97.82 in 12 patients and mean downward shift of 1.51 in 44 patients. Similarly, Verbesselt et al. [5] found mean FSIQ of 76 in 19 patients, corresponding to a downward shift of 1.6 SD. Although 22q11.2 CNVs appeared to shift the distribution to the left compared to the general population, they did not change its clinical features. 

While there was an average seven-point difference (0.5 SD) in core language scores favoring the children with 22q11.2Dup, the differences were not statistically significant. Differences were expected because duplications are generally associated with milder phenotypes compared to deletions [59]. In addition, parents reported more communication problems in children with 22q11.2DS than in children with 22q11.2Dup [13]. General communication concerns were reported in 47% of children with 22q11.2Dup and 79% of children with 22q11.2DS, whereas the current study found considerable language problems in 62% of children with 22q11.2Dup compared with 83% of children with 22q11.2DS. Therefore, results based on parental reports might reflect that parents of children with 22q11.2Dup may underestimate the language difficulties of their children. One might ask whether this finding is related to the fact that duplications are more often inherited and that affected parents might experience developmental issues and hence have difficulties with completing the questionnaires and/or correctly assessing the abilities of their children [60]. Nevertheless, many parents of children with the 22q11.2Dup were highly educated in the current study. More likely, discrepancies might be related to the subjective nature of indirect assessment methods. Therefore, indirect measurements such as questionnaires should be validated by direct assessment such as standardized language instruments to provide additional information about the true language capacities of the child. Another possible explanation for this discrepancy might be the fact that the CELF assesses structural and semantic language components, whereas the CCC-2 questionnaire screens speech, structural, semantic, and pragmatic language skills. Consequently, the lower reported proportion of communicative problems (47%) may be attributed to better speech or better pragmatic than structural and semantic language skills in children with 22q11.2Dup. 

Mean language scores of both CNV groups could be classified as within the range of mild–moderate language impairments. Within both CNV groups, similar scores were found for RLI and ELI. The presence of differences between receptive and expressive language in 22q11.2DS is the subject of debate in the literature. Pre-school children with 22q11.2DS often showed higher receptive than expressive skills, while there was more varied reporting in the literature on receptive–expressive discrepancies in the school-aged population [2,61,62,63,64]. The current results in the 22q11.2Dup group suggest that children with 22q11.2Dup might experience comparable receptive and expressive language challenges. Future longitudinal studies should clarify whether these similar receptive and expressive deficits are characteristic of the 22q11.2Dup population. Within the 22q11.2Dup group, children with speech-language delays in infancy obtained statistically significantly lower core language scores at primary school age than children without speech-language delays. These results might suggest that delayed milestones in infancy are indicative of language impairments in primary school. Nevertheless, these results should be interpreted with caution, since a history of speech-language delays did not always lead to core language impairments (*n* = 3, 17%) and others still developed language impairments without a history of speech-language delays (*n* = 2, 22%). 

Children with 22q11.2DS showed higher proportions of difficulties with subtests at the word level, such as Word definitions, Expressive Vocabulary, and Word Classes, whereas children with 22q11.2Dup predominantly demonstrated difficulties with subtests on the sentence level. However, difficulties with word-level subtests were not equally representative due to the smaller age range and accompanying smaller sample. Regarding the remaining subtests, the most common problems were found across the same subtests for children with 22q11.2 CNVs, but higher proportions of difficulties were found in the 22q11.2DS group. Therefore, children with 22q11.2 CNVs seemed to have deficits across several language domains, including both lexico-semantic problems (based on Concepts and Following Directions, Sentence Structure/Semantic Relations) and morpho-syntactic problems (based on Recalling Sentences and Formulated Sentences). Moreover, both language production and comprehension may be impaired in both CNV groups. Results for the expressive language composite based on Recalling Sentences and Formulated Sentences were in line with the core language scores. Higher proportions of difficulties with Concepts and Following Directions and Recalling Sentences might be related to impaired working memory, attention, and executive functioning in both groups of children, which has been previously described in 22q11.2DS [2,65,66,67,68,69,70]. Thus, qualitatively, children with 22q11.2Dup seemed to show lower proportions of language difficulties and higher language scores, with challenges across similar domains compared to children with 22q11.2DS.

Regarding confounding factors, only comorbid ASD, SES, and the specific CNV seemed to play a role in the language scores, although these effects did not remain statistically significant after correction for multiple testing. Children with ASD only performed worse in the 22q11.2DS group and children with high SES only showed higher scores in the 22q11.2DS group. Consequently, these factors did not seem to influence language outcomes in children with 22q11.2Dup. Qualitatively, the opposite pattern was shown for other variables, such as the observation of higher language scores in children with 22q11.2Dup and mild hearing loss or ASD. However, these differences cannot be generalized and should be interpreted with caution due to the small and imbalanced numbers. Moreover, differences for hearing loss and ASD were smaller than one standard deviation, and, thus, clinically less relevant. Future studies with more participants are needed to further delineate the impact of these and other potentially confounding factors. To conclude, qualitatively, children with 22q11.2Dup seemed to be in an intermediate position between the general population and age-matched children with 22q11.2DS, albeit with more overlap with the 22q11.2DS group. 

Language problems were observed both in patients with duplications proximal to LCR22-B, as well as in patients with more distal duplications, pointing toward downstream effects or the presence of multiple copy number sensitive loci for language development on chromosome 22q11. Minimal regions of overlap in patients with language problems included LCR22A-LCR22B, LCR22B-LCR22C and LCR22C-LCR22D. Woodward et al. [8] described thirteen atypical, nested duplications and investigated candidate genes within the LCR22B-LCR22D interval that might be associated with brain development and ASD traits. Based on gene expression in tissue of the nervous system, nine genes (ZNF74, KLHL22, MED15, PI4KA, SERPIND1, CRKL, AIFM3, SLC7A4, and BCRP) in the LCR22B-LCR22D interval were selected as candidate genes for these traits. Their potential link to language impairment in particular has not been established so far. Future studies with larger samples of typical and nested 22q11.2Dup, in addition to case-control variant burden studies in extensive cohorts of patients with severe language problems, are required to study the contribution of variation in genes on 22q11.2 to abnormal language development.

### Strengths, Limitations, and Future

To the best of our knowledge, this is the first study to measure language in children with 22q11.2Dup. By combining data from two cohort sites, a relatively large sample was obtained, increasing statistical power. However, data in CHOP were retrospectively collected, resulting in missing data for certain variables, such as information on inheritance pattern, SES, type of education, and the presence of NDD or mild hearing loss. The current sample size, although relatively large for these CNVs, might still prevent us from finding significant differences between 22q11.2 CNVs. Hence, future studies with larger samples are needed to confirm the current results. Another methodological limitation is the use of different versions of the same language test, which might lead to slightly different outcomes. Nevertheless, all CELF editions are constructed to measure the same language components and overall language ability and are recognized and used in clinical practice. Additionally, the self-constructed expressive composite showed similar results compared to the CLS, confirming its reliability and validity. Other strengths of the current study include the use of gold standard language assessments, strict inclusion and exclusion criteria, and comparisons with the normative sample. The large variability within the CNV groups points to the role played by other factors in language outcomes, such as the broader genetic background, IQ, hearing, ASD, ADHD, SES, which should be further elucidated in future studies. 

As the present study only characterized semantic and structural language skills, future studies should also delineate pragmatic skills and speech abilities through direct assessments, just as was established in 22q11.2DS [1,71]. All patients were diagnosed based on medical or developmental indications for genetic testing and only probands were included. Consequently, language outcomes may reflect the more severe end of the phenotypic spectrum due to ascertainment bias in this clinical cohort. It is likely that milder language impairments are present in those individuals identified with the 22q11.2Dup only following the diagnosis in their affected relative (e.g., a parent or sibling) as they may not have come to attention a priori with medical or developmental problems. Future studies should also include non-probands and affected relatives to obtain a complete picture of language skills in the 22q11.2Dup population. In addition, language should be interpreted in relation to the overall cognitive profile given the relationship between language and cognition. Hence, it would be an added value to perform intelligence testing prospectively in future research. Finally, the inclusion of larger samples and the collection of longitudinal data may lead to a more complete overview of the phenotypic spectrum of children with 22q11.2Dup across the lifespan. 

## 5. Conclusions

This is the first study to characterize language skills in children with 22q11.2Dup through direct language instruments, in relation to typically developing peers in the general population and age-matched children with 22q11.2DS. Considerable language difficulties were found in a high proportion of children with 22q11.2 CNVs. Therefore, as in 22q11.2DS, regular follow-up of language development in children with 22q11.2Dup is advised. Early screening and characterization of language skills in 22q11.2Dup are recommended to identify children who qualify for educational support in school or speech-language therapy through a rehabilitation center or private practice. Finally, as in 22q11.2DS, adapted treatment is suggested to support and improve language skills and to reduce potential long-term influence of language and communicative deficits [1].

## Figures and Tables

**Figure 1 genes-14-00679-f001:**
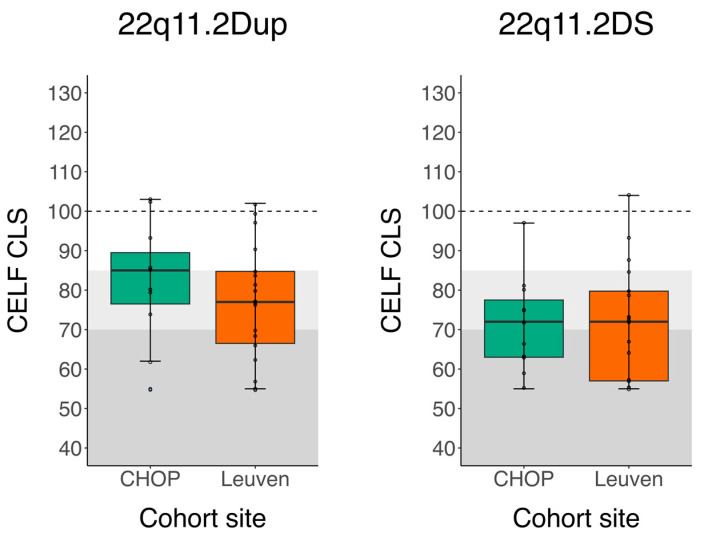
Boxplots for CELF CLS composite scores across CNV groups and cohort sites. The dotted lines show norm group averages. The gray zones indicate the severity of the problems; the darker the gray, the more severe the difficulties: mild–moderate = light gray zone and severe = darker gray zone, based on clinical cut-off scores for the CELF. Abbreviations. CELF, Clinical Evaluation of Language Fundamentals (norm group average = 100, cut-off: <85 = mild–moderate (pc 16), <70 = severe (pc 2).

**Figure 2 genes-14-00679-f002:**
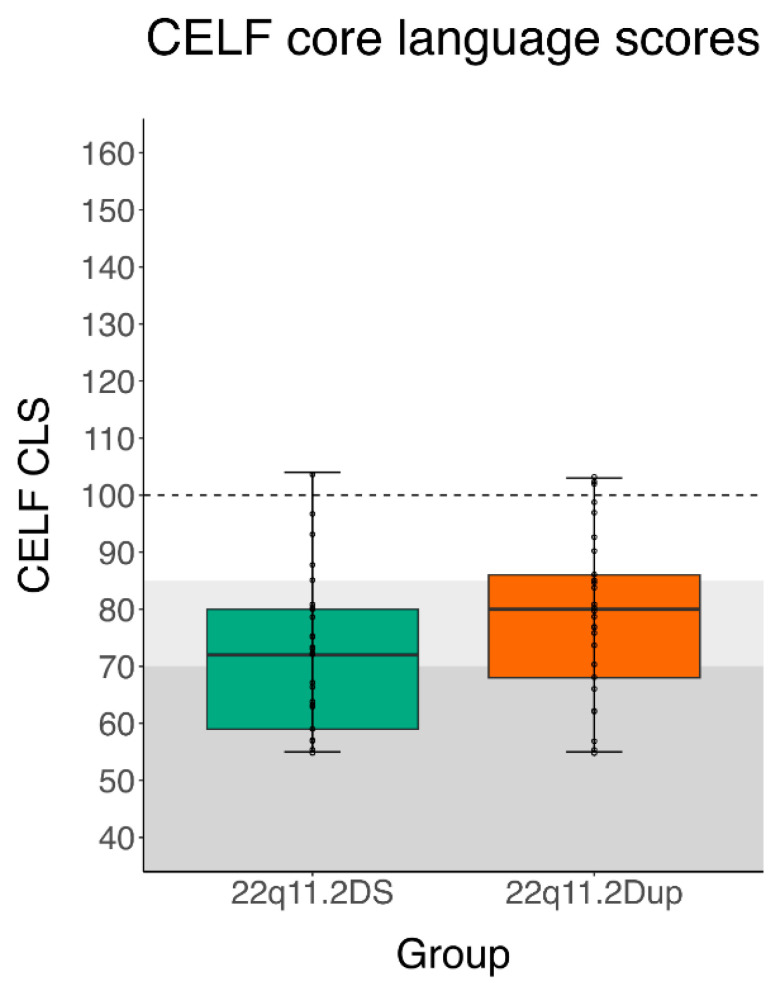
Boxplots for CELF Core Language Scores across groups (22q11.2DS and 22q11.2Dup).

**Figure 3 genes-14-00679-f003:**
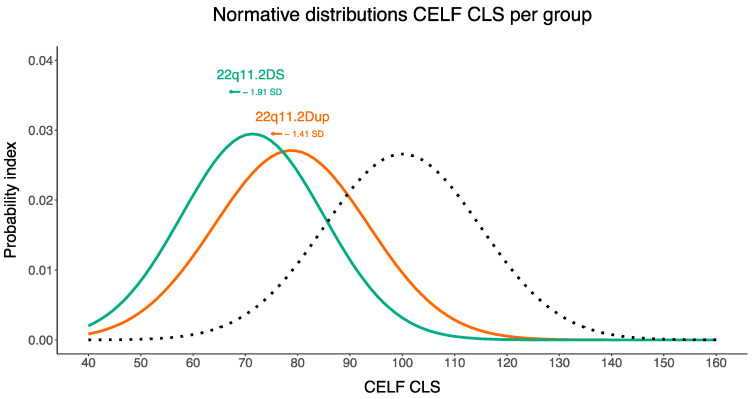
Normative distributions of CELF Core Language Scores across groups (22q11.2DS and 22q11.2Dup). SD, standard deviation. The dashed line illustrates the normative distribution of the norm group (mean = 100, SD = 15). SD shifts are calculated in relation to the normative sample.

**Table 1 genes-14-00679-t001:** Demographic and clinical features across groups.

	22q11.2DS		22q11.2Dup	
Native Language (Country of Residence)	Dutch (Belgium)	English (USA)	Dutch (Belgium)	English (USA)
Sample Size (*n*)	18	11	18	11
Sex (*n*, %) Male Female	12 (67%)	8 (73%)	10 (56%)	7 (64%)
	6 (33%)	3 (27%)	8 (44%)	4 (36%)
Chronological age (yrs.mo) Average (SD) Median Range				
	10.10 (2.8)	9.1 (2.5)	10.10 (2.7)	9.1 (2.5)
	11.3	8.2	11.2	8.3
	6.7–15.2	6.4–13.2	6.11–15.5	6.4–13.3
Type of education (*n*, %) Special education Regular education Regular with assistance Homeschool Unknown				
	11 (61%) 6 (33%)	2 (18%) 0 (0%)	7 (39%) 6 (33%)	3 (27%) 2 (18%)
	1 (6%) 0 (0%) 0 (0%)	6 (55%) 0 (0%) 3 (27%)	5 (28%) 0 (0%) 0 (0%)	3 (27%) 2 (18%) 1 (10%)
SES High Middle Low Unknown	8 (44%) 10 (56%) 0 (0%) 0 (0%)	5 (45%) 5 (45%) 0 (0%) 1 (10%)	11 (61%) 6 (33%) 1 (6%) 0 (0%)	2 (18%) 4 (36%) 1 (10%) 4 (36%)
Speech-language delays (*n*, %)	18/18 (100%)	4/5 (80%)	14/18 (78%)	4/9 (44%)
Speech-language therapy (*n*, %)	18/18 (100%)	5/5 (100%)	15/18 (83%)	9/11 (82%)
Formal NDD diagnoses (*n*, %)				
ASD	7/18 (39%)	0/4 (0%)	2/18 (11%)	4/11 (36%)
ADHD	3/18 (17%)	0/4 (0%)	3/18 (17%)	3/10 (30%)
Inheritance pattern (*n*, %) De novo Inherited: Maternally inherited Paternally inherited Unknown	17/18 (94%)	4/11 (36%)	8/18 (44%)	1/11 (10%)
	1/18 (6%) 1/1 (100%) 0/1 (0%)	0/11 (0%) 0/0 (0%) 0/0 (0%)	7/18 (39%) 2/7 (29%) 5/7 (71%)	5/11 (54%) 2/5 (40%) 3/5 (60%)
	0/18 (0%)	7/11 (64%)	3/18 (17%)	5/11 (54%)
Medical issues (*n*, %)				
CHD	10/18 (56%)	7/11 (64%)	2/18 (11%)	1/11 (10%)
Palatal defects	11/18 (61%)	11/11 (100%)	4/18 (22%)	0/11 (0%)
Mild HL	9/18 (50%)	1/6 (17%)	2/18 (11%)	3/11 (27%)

Abbreviations: SES, socioeconomic status; NDD, neurodevelopmental disorders; ASD, autism spectrum disorder; ADHD, attention-deficit/hyperactivity disorder; CHD, congenital heart defects; mild HL, hearing loss (≥ 20 dB *HL* and ≤40 dB *HL*).

**Table 2 genes-14-00679-t002:** Mean composite results on CELF across CNV groups and cohort sites.

	22q11.2Dup Dutch (*n* = 18)	22q11.2Dup English (*n* = 11)	Statistical Outcomes Independent *t*-Test	22q11.2DS Dutch (*n* = 18)	22q11.2DS English (*n* = 11)	Statistical Outcomes Independent *t*-Test
CELF CLS Mean (SD) Range 95% Confidence interval	76.72 (14.67) 55.00–102.00 69.43–84.02	82.18 (14.84) 55.00–103.00 72.21–92.15	*t* = 0.968 *p* = 0.342	71.33 (14.74) 55.00–104.00 64.00–78.66	71.46 (12.00) 55.00–97.00 63.39–79.52	*t* = 0.023 *p* = 0.982
CELF RS + FS Mean (SD) Range 95% Confidence interval	11.83 (6.19) 2.00–24.00 8.76–14.91	13.09 (5.89) 4.00–22.00 9.13–17.05	*t* = 0.541 *p* = 0.593	10.28 (5.42) 2.00–20.00 7.58–12.97	11.46 (3.91) 8.00–20.00 8.83–14.08	*W* = 107.000 *p* = 0.734

Statistical outcomes: *p*-value; *α* = 0.05; *α* after Bonferroni correction = 0.025; *t*-value or *W*-value. CELF, Clinical Evaluation of Language Fundamentals (norm group average = 100, cut-off: <85: mild–moderate, <70: severe); CLS, Core Language Score; RS, Recalling Sentences; FS, Formulated Sentences (norm group average = 20).

**Table 3 genes-14-00679-t003:** Cross-CNV comparisons across CELF CLS and expressive composite scores.

	Scores	22q11.2DS	22q11.2Dup	Statistical Outcomes Paired *t*-Test
CHOP + Leuven (*n* = 29)	CELF CLS Mean (SD) Range 95% Confidence interval	71.38 (13.54) 55.00–104.00 66.23–76.53	78.79 (14.72) 55.00–103.00 73.19–84.39	*t* = 1.982 *p* = 0.057 *d* = 0.368
	CELF RS + FS Mean (SD) Range 95% Confidence interval	10.72 (4.86) 2.00–20.00 8.87–12.57	12.31 (6.00) 2.00–24.00 10.03–14.59	*t* = 1.219 *p* = 0.233 *d* = 0.226
Leuven (*n* = 18)	CELF CLS Mean (SD) Range 95% Confidence interval	71.33 (14.74) 55.00–104.00 64.00–78.66	76.72 (14.67) 55.00–102.00 69.43–84.02	*t* = 1.153 *p* = 0.265 *d* = 0.272
	CELF RS + FS Mean (SD) Range 95% Confidence interval	10.28 (5.42) 2.00–20.00 7.58–12.97	11.83 (6.19) 2.00–24.00 8.76–14.91	*t* = 0.906 *p* = 0.378 *d* = 0.214
CHOP (*n* = 11)	CELF CLS Mean (SD) Range 95% Confidence interval	71.46 (12.00) 55.00–97.00 63.39–79.52	82.18 (14.84) 55.00–103.00 72.21–92.15	*t* = 1.681 *p* = 0.124 *d* = 0.507
	CELF RS + FS Mean (SD) Range 95% Confidence interval	11.46 (3.91) 8.00–20.00 8.83–14.08	13.09 (5.89) 4.00–22.00 9.13–17.05	*t* = 0.790 *p* = 0.448 *d* = 0.238

Statistical outcomes: *p*-value; *α* = 0.05; *α* after Bonferroni correction = 0.008; *t*-value; Cohen’s *d* as effect size. CLS, Core Language Score (norm group average = 100, cut-off: <85: mild–moderate, <70: severe); RS, Recalling Sentences; FS, Formulated Sentences (norm group average = 20).

**Table 4 genes-14-00679-t004:** Proportions of children with difficulties across composite and subtest scores on CELF.

		22q11.2DS	22q11.2Dup
Composite scores	CELF CLS (< −1 SD & < −2 SD) Mild–moderate < −1 SD Severe < −2 SD	24/29 (83%) 11/29 (38%) 13/29 (45%)	18/29 (62%) 10/29 (34%) 8/29 (28%)
	CELF RLI	10/12 (83%) *	19/27 (70%)
	CELF ELI	11/12 (92%)	18/28 (64%)
Subtest scores Receptive	CFD	21/29 (72%)	17/27 (63%)
	SS / SR	14/27 (52%)	15/28 (54%)
	WC	11/19 (58%)	10/29 (34%)
Subtest scores Expressive	RS	16/29 (55%)	19/29 (66%)
	FS	22/29 (76%)	15/29 (52%)
	WS (5.0–8.11 years)	5/10 (50%)	6/12 (50%)
	EV (5.0–9.11 years)	4/7 (57%)	2/7 (28%)
	WD (≥10.00 years)	9/10 (90%)	6/12 (50%)

* Available data vary by subtest due to different age ranges of specific subtests or missing data. CLS, Core Language Score; RLI, Receptive Language Index; ELI, Expressive Language Index (cut-off: <85: mild–moderate, <70: severe); CFD, Concepts and Following Directions; RS, Recalling Sentences; FS, Formulated Sentences; WS, Word Structure; SS, Sentence Structure (5.0–8.11 years); SR, Semantic Relations (≥9.0 years); WC, Word Classes; EV, Expressive Vocabulary; WD, Word Definitions (cut-off: <7: mild–moderate problems; <4: severe problems).

## Data Availability

The datasets used and/or analyzed during the current study are available from the corresponding author upon reasonable request.

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
