# Peer review of "Language Profiles of School-Aged Children with 22q11.2 Copy Number Variants"

_genes, 2023, doi:10.3390/genes14030679_

Round 1
Reviewer 1 Report
A brief summary
The aim of the paper is to study language profiles in school-aged children with 22q11.2 Copy Number Variants and to compare language abilities in children with 22q11.2 deletion (22q11.2DS) and 22q11.2 duplication (22q11.2dup). Further research questions were to study if the spoken languages (Dutch vs English) and accompanying cultural differences play a role in the outcome of language profiles and to determine the impact of several confounding factors on language outcomes. The strengths of the study is that authors have used a standardized language test to compare language outcomes in groups of 22q11.2DS and 22q11.2dup as this have not been studied earlier. They have shown that children with 22q11.2 CNV have more language problems compared with the general population.
General concept comments
Areas of weakness
I suggest that you formulate the research questions more clearly. Some of them are not formulated as questions (or as hypothesis). Now it is written that the research questions are….
The Dutch group is investigated prospectively and the group from CHOP retrospectively and the latter group (CHOP) is investigated with different versions of the standardized language test CELF. Further, it seems to lack information on some of the children from CHOP on neurodevelopmental diagnoses, hearing and information on speech-language delays and speech-language therapy. The lack of data is not clearly stated.
Based on the data, including heterogeneous small groups, I can’t see how it would be possible to answer the question ‘Did the spoken languages (Dutch versus English) and accompanying cultural differences play a role in the outcome of language profiles in children with 22q11.2 CNVs?’ Other random effects of differences between the groups (for example IQ) could have a higher impact on language outcome than native language or cultural differences. I suggest that you rephrase this question. You already present results on cohort site differences, which is fine. Adapt the question after the results.
I suggest that you don’t compare across subgroups as these include few individuals, there are differences in test procedure and there are several confounding factors that are difficult to control for. For example on page 8, the first paragraph. “Within the 22q11.2DS group, children with speech-language delays (n=22) showed mean CLS of 70.55 compared to CLS of 97 in one child without speech-language delays”. This type of comparisons should be removed. The same for all the comparisons with small subgroups with different confounding factors. In Table 1 we have descriptive figures of the clinical features/confounding factors. That is good. However, there seems to be missing data on several of the features (for example NDD). This together with small subgroups makes all the comparisons on page 10 (the second and third paragraph) unsure. I suggest again that you delete all these comparisons of very small subgroups. This should be studied in larger future studies.
Interpretation of language outcome – on page 7, the last paragraph, it is stated that no statistically significant differences in CLS between 22q11.2DS and 22q11.2Dup was revealed…..subtest scores revealed similar distributions across all subtests, suggesting that children with 22q11.2DS and children with 22q11.2Dup in this sample have comparable language skills. In the abstract it is stated “Mean language skills were better in the children with the 22q11.2Dup compared to those with 22q11.2DS, although these findings did not reach statistical significance. Please, do these statements comparable. In this study there is no statistical significant difference. With larger samples it might have been a difference. This can be discussed in the discussion.
Specific comments
Table 1 – it should be clearly stated that you have missing data on several of the clinical features (type of education, speech-language delay, speech-language therapy, NDD, maternal/paternal inheritance patterns, hearing loss). The inheritance pattern is difficult to understand. If think the share of ‘de novo’, ‘inherited’ and ‘unknown’ are reported as an entity. Share of ‘maternal’ and ‘paternal’ are reported as a separate entity, but are between ‘inherited’ and ‘unknown’ in the table. Please move maternal and paternal.
Table 4 – Should it be <-1 SD instead of <1SD? Mild-moderate should it be <1SD->2SD? On the first row in the table CELF CLS (<1SD & <2SD), can it be written differently as all are included in <1SD?
In the top paragraph on page 8, CLS in children with delayed speech-language milestones in infancy are reported. This is coming back in the end of the first paragraph on page 9. I suggest these two parts are moved together and the score from the subtest are in the end of the paragraph.
In the beginning of the Discussion differences between participants from the two sites are discussed. You could also discuss how representative your groups ‘DS’ and ‘Dup’ are in comparison to larger studies. For example the male/female distribution is rather even in larger studies of 22q11.2DS while it is a higher proportion of males in the current study. CHDs and palatal defects can also be compared with the frequency in larger studies.
In the discussion on differences between parent reported problems using CCC-2 and CELF, it might be better to compare the outcomes with the specific language sub scales on CCC-2, as speech problems are very common especially in 22q11.2DS. These difficulties are not part of the examination in CELF.
Conclusions – do we really have evidence for how to treat and if treatment optimize language skills in this group?
Author Response
Please see the attachment. The correct attachment is 'author-coverletter-27006618.v2.docx'

Reviewer 2 Report
The study was well designed and conducted and the paper is very well written.
Although the focus of the paper is not on genetic aspects, considering that this is a Genetics Journal, it would be interesting to include some discussion about candidate genes, in this region, for speech issues. Interestingly, in three patients with 22q11 duplications (LCRs C-D, B-F, B-C) the most proximal region, which includes TBX1 gene, is not duplicated. Do these patients present language impairment? Could the author establish a minor region of overlap and point to candidate genes for this phenotype?
Also, some discussion about dysregulation of genes in this region, regardless of being deleted or duplicated, which could impact language phenotype, could be added
Author Response
Please see the attachment. The correct attachment is 'author-coverletter-27188544.v2.docx'

Reviewer 3 Report
The authors analyzed language skills in children with 22q11.2Dup and 22q11.2DS and found considerable language difficulties in a high proportion of children with 22q11.2 CNVs. The population is well-defined, and the methodology is described in detail. The strengths and limitations of the study have been thoroughly discussed, and a vision of future research is given. There is only one minor point:
Last Paragraph of the Result section 3.4 Influence of Confounding Factors, the sentence “There is only one child with an inherited deletion, with lower CLS compared…” I would suggest correcting this sentence to be in the Past Tense to keep reporting the results consistently: “There was only one…”
Round 2
Reviewer 1 Report
Thank you for your answers and for your revision of the manuscript, which is substantially improved.
I have just one additional suggestion. You have added genotype-phenotype comparisons in the result section. I suggest this is added to the research questions as well.